# Nanocellulose Coating on Kraft Paper

Elaine Cristina Lengowski [1], Eraldo Antonio Bonfatti Júnior [2,*], Leonardo Coelho Simon [3],
Graciela Inês Bolzon de Muniz [2], Alan Sulato de Andrade [2], Aleffe Neves Leite [1]
and Emilly Laize Souza de Miranda Leite [1]

1 Faculty of Forestry Engineering, Federal University of Mato Grosso, Cuiabá 78060-900, Brazil;
elainelengowski@gmail.com (E.C.L.); aleffe.leite18@gmail.com (A.N.L.);
emilly.sml@hotmail.com (E.L.S.d.M.L.)
2 Department of Engineering and Forestry Technology, Federal University of Paraná, Curitiba 80210-170, Brazil;
graciela.ufpr@gmail.com (G.I.B.d.M.); alansulato@gmail.com (A.S.d.A.)
3 Department of Chemical Engineering, University of Waterloo, Waterloo, ON N2L 3G1, Canada;
leonardo.simon@uwaterloo.ca
* Correspondence: bonfattieraldo@gmail.com; Tel.: +55-41-998-302-630

**Abstract:** Paper is a biodegradable material, but in food packaging, its hygroscopicity and porosity can cause food contamination due to the exchange of gasses and liquids with the environment. Therefore, it is important to use biodegradable materials for paper coatings, such as nanocellulose, which is chemically compatible with paper but less hygroscopic. The objective of this study was to evaluate the effectiveness of nanofibrillated cellulose (NFC) as a paper coating. NFC produced from bleached eucalyptus pulp was used as a coating on kraft paper sheets produced from Pinus pulp. To prepare the coating, two thicknesses of wet nanocellulose (1 mm and 2 mm) were tested, and two nanocellulose films made with the same thicknesses were evaluated. The morphological, physical, mechanical, and thermal properties of the composites were investigated. The presence of NFC improved the surface of the paper by filling the pores; consequently, the density and barrier properties were also improved. All mechanical properties were improved, with the highest increases observed for bursting and tensile strength; however, the films showed low bursting index values and null values for the tearing index. The thermal stability of the paper with NFC coatings met the minimum requirements for food packaging.

**Keywords:** nanotechnology; food packaging; waterproof; thermal stability; air permeance

## 1. Introduction

Food packaging can be composed of metal, plastic, glass, or paper [1]. Plastics have a large share of the food packaging market, and there are many examples of foods traditionally packaged in glass containers that are found only in plastic packaging, such as pasteurized milk [2]. The primary plastic materials used in the production of food packaging are low-density polyethylene (LDPE), linear low-density polyethylene (LLDPR), high-density polyethylene (HDPE), polypropylene (PP), ethylene-vinyl acetate (EVA), ethylene-vinyl alcohol (EVOH), ethylene acrylic acid copolymer (EAA), polystyrene (PS), polyvinyl chloride (PVC), polyvinylidene chloride (PVDC), polyethylene terephthalate (PET), polyethylene naphthalate (PEN), polyamide (PA), and polycarbonate (PC) [3].

The success of plastic is due to its low cost, lightness, flexibility, durability, corrosion resistance, and high machinability [4], and its barrier properties are advantageous for food packaging [3,5]. In 2020, plastic production was 363 million tons, making up a market of USD 580 billion [6]; by 2050, it is expected to reach 500 million tons per year [4]. Despite the advantages of this material, it is important to emphasize the problem of plastic pollution, which increases simultaneously with plastic production [6,7].

Despite the availability of recycling technologies for plastic, only 9% of discarded plastic is recycled, while the remainder is incinerated (12%), deposited in the environment, or

disposed of in landfills (79%) [8]. The most used plastics are not biodegradable, remaining in the environment for a long time, often centuries [5,9]. This promotes the accumulation of plastics (plastic nanomaterials and plastic micromaterials) in the environment and, consequently, in living organisms through the food chain [10]. Plastic accumulation can cause lung, intestinal, kidney, cardiac, and liver diseases, in addition to genetic mutations, nervous system degradation, and decreased fertility in humans and animals [11–15].

Limiting the use of plastic is essential for natural conservation and environmental sustainability [7]. For food packaging, cellulosic materials are suitable alternatives to plastic [3]. Paper, the primary cellulosic material, is utilized in the food industry as bags, cartridges, cardboard boxes [3,16], and, more recently, bottles [17,18].

Paper is essentially a network of natural cellulosic fibers, usually derived from wood [19,20]. To produce paper, it is necessary to individualize the wood fibers through chemical or mechanical processes. The obtained fiber solution is dried in appropriate machines to form paper [20]. Paper can be obtained from short-fibered hardwoods, producing highly malleable paper with little mechanical resistance, making it suitable for printing, writing, and tissue [20,21]. Softwood, on the other hand, has longer fibers that provide high mechanical strength to paper, which is suitable for packaging [20].

The kraft pulping process involves solubilizing wood lignin through an alkaline solution of sodium hydroxide (NaOH) and sodium sulfide ($Na_2S$) at high temperatures and pressures [20]. This process is responsible for more than 90% of all papers produced worldwide [22] because it provides good mechanical strength [20]. Kraft paper refers to the paper produced through this process, which is normally made from unbleached coniferous wood, with grammage ranging from 50 to 300 $g/m^2$ [3]. This type of paper has good mechanical properties and good moisture resistance, but it is still inferior to some plastics [23].

For a more effective use of paper in food packaging production, strategies that can improve its mechanical strength, barrier properties, and thermal properties are needed [20]. One such strategy is the use of coatings, which form an extra layer and provide the paper with adequate characteristics [24]. However, the coating material must also be of natural origin, biodegradable, and non-toxic, like the paper itself [3,25,26]. Polysaccharides such as starch, nanocellulose, and chitosan are some examples that meet these requirements [26,27].

Nanocellulose refers to all cellulosic materials with at least one dimension on the nanometric scale [20], including nanocellulose crystals (CNC), microfibrillated cellulose (CMF), and nanofibrillated cellulose (NFC) [28]. Because nanocellulose is obtained from plant biomass, it is chemically compatible with paper [20], and its application does not alter the paper's characteristics, reduce its biodegradability, or make it more dangerous to human health [20,27,29]. NFC exhibits hydrophobicity [24], and when applied as a coating on paper, it enhances the paper's barrier properties, such as its permeability to air, gas, oil, and grease [29], and improves its thermal and mechanical properties [3,20,27], making it a more suitable material for food packaging [3].

Nanocellulose is not the only renewable compound that can improve the barrier properties of paper; starch, chitosan, alginate, and polylactic acid (PLA) are useful in this role [27]. Furthermore, modifying the paper surface is a viable alternative, if the products of these modifications are non-toxic [3].

Plastic remains one of the most substantial and challenging innovations to replace in the 21st century [4]. For food packaging, paper alone cannot replace it completely, but with the use of a nanocellulose coating, the use of paper can be expanded to provide adequate protection to various foods [3]. In this study, the performance of kraft paper with two coating thicknesses (1 and 2 mm), as well as nanocellulose films made with the same thicknesses of the coatings of papers, was evaluated. Its morphological characteristics, physical properties, barrier properties, and mechanical properties were evaluated.

## 2. Materials and Methods

### 2.1. Materials

The paper used in this study was produced in the laboratory from unbleached *Pinus* spp. kraft pulp on a Rapid-Köethen paper machine (Regmed, Osasco, Brazil) with a final grammage of 60 g/m$^2$ [30]. Industrial eucalyptus kraft pulp bleached with an elemental chlorine-free (ECF) bleaching sequence was used to produce NFC. An aqueous suspension of the pulp with 1.3% consistency was prepared by homogenization in a blender. Subsequently, this suspension was processed with a Masuko Sangyo Super Masscolloider MKCA6-2J (Masuko Sangyo, Kawaguchi, Japan) to obtain a gel-like suspension in water [31]. To obtain the cellulose nanofibrils, a rotation frequency of 1500 rpm and 10 passes through the mill were adopted.

### 2.2. Coating of Papers and Films Production

The NFC coatings were deposited on the kraft paper through the laminar method on a flat surface at 1 and 2 mm of NFC suspension thickness and previously dried in an oven at 50 °C for 20 min, then dried in a Rapid-Köethen sheet dryer [30] at 90 °C and a pressure of 80 kPA until completely dry (8% humidity). Films without paper were also made with NFC suspension thicknesses of 1 mm and 2 mm, following the same steps as for the coatings.

In total, five treatments were analyzed in 10 replications and named as follows:

- Uncoated kraft paper (sample called control KP);
- Kraft paper with deposition of 1-mm of NFC suspension (sample called K1);
- Kraft paper with deposition of 2-mm of NFC suspension (sample called K2);
- NFC 1-mm film (sample called F1);
- NFC 2-mm film (sample called F2).

Before all analyses described below, the papers, paper coatings, and films were previously packed in a climate-controlled room with a temperature of 23 ± 2 °C and a relative humidity of 50 ± 2% [32].

### 2.3. Morphological Analysis

A morphological analysis of the cellulose nanofibrils (NFC) was performed via transmission electron microscopy (TEM) using a JEOL JEM 1200EXII Transmission Electron Microscope (600×) (Jeol, Peabody, MA, USA). In this analysis, the NFC suspension was diluted to 5 ppm in deionized water and dripped onto the surface of a 400-mesh screen. The samples were then allowed to dry at room temperature. To check the size of the fibrils, three diameter measurements were taken for each NFC micrograph.

To analyze the morphology of the papers with and without the application of the NFC coating, SEM images were obtained using a HITACHI scanning microscope (model TM-1000) (HITACHI, Tokyo, Japan) and a Philips microscope (model XL 30 series) (Philips, Eindhoven, The Netherlands). For this analysis, the samples were metallized using an electron beam to better identify their structures.

### 2.4. Physical Tests

The physical characterization of the papers and films was performed by considering their apparent density [33] and water absorption using the Cobb method [34] and the airflow resistance using the Gurley method [35]. In the case of the water absorption of the NFC-coated papers, both sides of the papers were evaluated (coating side and paper side).

In addition, surface wettability was characterized by determining the apparent contact angles of the samples. A Data Physics OCA goniometer was used to apply the sessile drop method at room temperature (20 ± 2 °C). Three droplets of deionized water (5 μL) and three droplets of glycerol (5 μL) were deposited in three different samples, totaling nine replicates for each treatment and solvent. To ascertain the wettability kinetics, measurements were performed at the moment of droplet contact with the sample (0) and after 5, 15, and 30 s of

contact between the droplet and sample surface. In this analysis, both sides of the coated paper were evaluated.

The nanocellulose-coated papers had their grammage determined before and after coating. Based on these values and the apparent density of the films, the dry thickness of the coating was determined.

### 2.5. Mechanical Tests

Tensile strength was determined with an adaptation of TAPPI T 494 om-13 [36], where the distance between the claws was 50 mm. This analysis was performed using a dynamometer. The index was calculated using the relationship between tensile strength and grammage. The burst resistance was measured according to TAPPI T 403 om-15 [37]. It is expressed in kPa, and its index is calculated as the ratio between the breaking strength and grammage. The tear strength was determined using the T 414 om-98 [38] standard of the Elmendorf Pendulum equipment and expressed in mN. Its index was calculated by the ratio between tear strength and grammage, expressed in $mN \cdot m^2/g$.

### 2.6. Thermal Stability

A thermogravimetric analysis (TGA) of the nanocellulose was performed using a Setaram Setsys Evolution TGA/DSC (Setaram, Lyon, France). The analyses were performed under an argon gas atmosphere at a constant flow rate of $40 \text{ mL} \cdot \text{min}^{-1}$, using approximately 5 mg of the sample. The ramp adopted was 10 °C per minute and the temperature ranged from 35 °C to 650 °C. A thermogravimetric (TG) curve was obtained to evaluate mass loss as a function of temperature. The first derivative of the mass loss was used to generate the second curve (DTG) to determine the onset, maximum, and end set temperatures for thermal degradation. The TG curves were used to calculate the mass loss in the following temperature ranges: 100–200 °C, 200–300 °C, 300–400 °C, 400–500 °C, and 500–600 °C. The initial, maximum, and final thermal degradation temperatures were determined.

### 2.7. Statistical Analysis

The following tests were performed for the statistical analysis: the Grubbs test for outliers, the Shapiro–Wilk test for data normality, Levene's test for homogeneity of variance, and the analysis of variance (ANOVA). Tukey's mean comparison test was performed when the equality hypothesis was rejected. All tests were performed using the Statgraphics Centurion XV Program (Statgraphics Technologies Inc., The Plains, VA, USA) at a 95% probability.

## 3. Results

### 3.1. Morphological Analysis

Refining promotes the breakdown of cellulosic fibers into fibers to increase the contact area and, consequently, the interlacing capacity, which is a fundamental condition for paper formation [39]. The fibers from *Pinus* sp. had micrometer-scale diameters (Figure 1a), but the refining process employed resulted in a small fibrillation of the wall (Figure 1b). The diameters found for the pine tracheids varied between 35 and 45 μm, with an average of 40 μm. After refining, the fibers underwent internal and external fibrillation, delaminating the outer layers of the cell wall (the P and S1 layers) and exposing the S2 layer [40] to produce nanometric elements on the fiber surface.

The mechanical defibrillation process resulted in cell wall defibrillation, producing NFCs for the bleached pulp. It was possible to verify the delamination in the outermost layers of the cell wall (Figure 2a), starting with the primary wall and then the secondary wall layers S1, S2, and S3 [40]. The measurement of the dimensions of the NFCs through micrographs revealed that the bleached NFC had an average diameter of 22.28 nm and a length of several micrometers (Figure 2b).

The deposition of NFC on paper reduces its porosity, as these nanomaterials occupy the empty spaces between the fibers [41]. With a larger surface area, NFC provides a denser and

more homogeneous surface. Surface quality is an important criterion for food packaging, as it is one of the characteristics that determine the barrier properties [3,20,24,27,29]. The reduction of empty spaces in paper can be considered a significant advancement for the use of this material in food packaging because, with lower porosity, barrier properties tend to improve [42]. Pore filling occurred in both treatments K1 (Figure 3a,b) and K2 (Figure 3c,d).

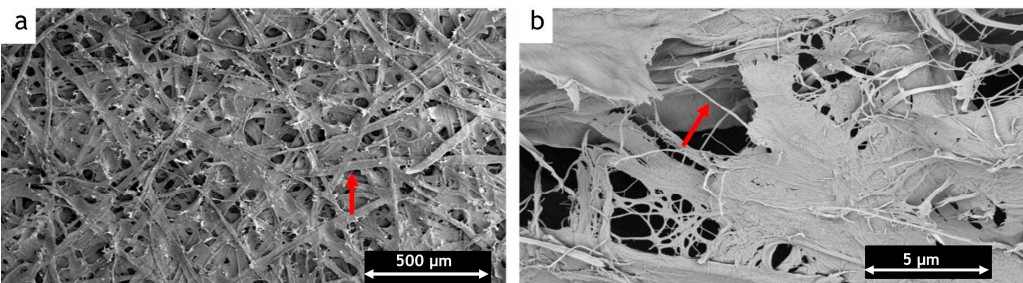

**Figure 1.** SEM images of kraft papers used for nanocellulose deposition. (**a**) Image with magnification 100× highlighting the fibers that form the paper; (**b**) image with magnification 10,000× highlighting the nanometric fibrils formed by refining.

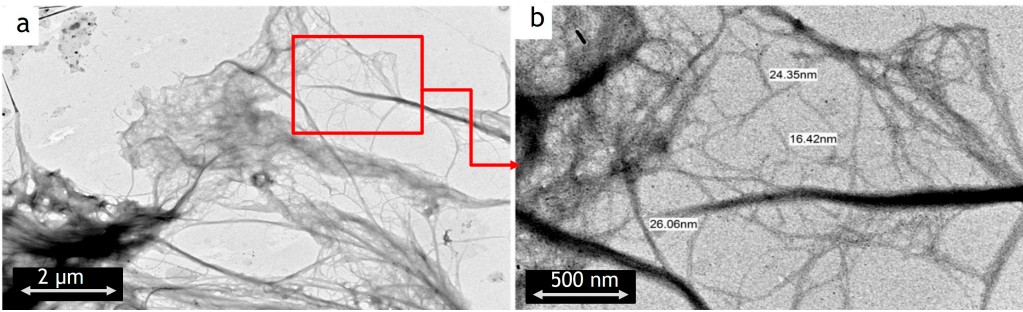

**Figure 2.** Transmission electron microscopy. (**a**) NFC produced with bleached *Eucalyptus* sp. pulp; (**b**) network appearance formed and dimensions after the fibrillation process.

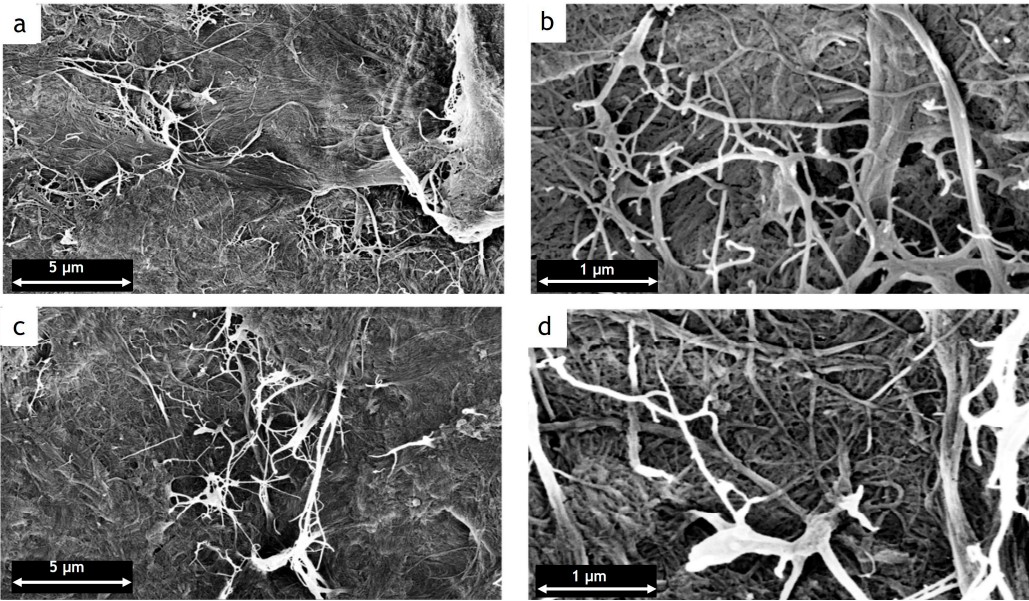

**Figure 3.** SEM images of kraft papers after deposition of the NCF coating. (**a**) Image with magnification 10,000× of the paper with 1mm NFC coating; (**b**) image with magnification 50,000× of the paper with 1mm NFC coating; (**c**) image with magnification 10,000× of the paper with 2 mm NFC coating; (**d**) image with magnification 50,000× of the paper with 2 mm NFC coating.

### 3.2. Physical Tests

The application of NFC coatings on paper resulted in a significant increase in their apparent density (Table 1). Both coatings increased the paper density without demonstrating statistical differences between the two treatments. A similar behavior was observed when the films were analyzed, both of which had statistically equal apparent densities. The papers with an NFC coating had an apparent density increase, which occurred because of the larger contact surface of the NFC and the greater interaction and compaction between them [43]. A denser surface of a material is related to its porosity and is important for improving its barrier properties [43,44].

**Table 1.** Mean values of apparent density and water absorption by Cobb test.

| Sample | Apparent Density (g/cm$^3$) | Water Absorption (g/m$^2$) |
|---|---|---|
| KP | 0.44 a $^{(1.73)}$ | 133.15 c $^{(3.90)}$ |
| K1cs | 0.59 ab $^{(11.33)}$ | 55.82 b $^{(1.43)}$ |
| K1 ps | | 192.82 d $^{(5.46)}$ |
| K2 cs | 0.63 bc $^{(2.12)}$ | 60.55 b $^{(4.16)}$ |
| K2 ps | | 195.86 d $^{(2.40)}$ |
| F1 | 0.73 cd $^{(8.58)}$ | 34.23 a $^{(5.26)}$ |
| F2 | 0.86 d $^{(13.96)}$ | 40.76 a $^{(7.44)}$ |

cs: coating side. ps: paper side. Means followed by the same letters indicate no statistical differences according to the Tukey test at 95% probability. Values in parentheses refer to the coefficient of variation in percent.

The deposition of the 1mm coating of the wet nanocellulose suspension increased the paper grammage by 20 g/m$^2$, while with the deposition of the 2mm coating, there was an increase of 28 g/m$^2$ in the paper grammage. As a result of this increase in weight, the consistency of the nanocellulose solution, and the apparent density of the coatings, the dry thickness of the paper coatings was 25 μm for K1 and 31 μm for K2. The non-proportional increase in dry thickness in relation to the wet thickness deposited is due to the nanocellulose suspension permeating the empty spaces that form between the cellulose fibers (Figure 1) during its deposition and drying, making the surface densification possible (Figure 3). Once these spaces are filled in the paper, the coating begins to form above the fibers.

Deposition methods that do not use pressure do not provide adequate control over the coating thickness [29]; in the present work, there is no evidence that the pressurized paper-forming machine is the solution to controlling the final thickness of the coating. On an industrial scale, calendaring can be used to homogenize the thickness [19].

The water absorption of the samples with nanocellulose coatings was significantly reduced compared to that of kraft paper without a coating (Table 1). The decrease in water absorption was 58% lower after the deposition of the 1-mm coating and 55% lower for the sample with the 2-mm coating deposition, and no statistical difference was found between the two surface deposition thicknesses (K1 and K2).

There was a considerable increase in the water adsorption on the surface, contrary to the surface deposition, where the absorption of the KP was 133.15 g/m$^2$. After the deposition, the water absorption increased to 192.82 g/m$^2$ for the K1 treatment and 195.86 g/m$^2$ for the K2 treatment. The greater surface tension present in NCF and the greater availability of hydroxyl groups indicate that more water is retained when exposed to the surface opposite the deposition [43,45]. The absorption of the nanostructured films is significantly lower than that of the sheet produced only with pulp (KP) (on average, 37.5 g/m$^3$ compared to 133.15 g/m$^2$), which is 3.5 times higher than the nanostructured films. This result corresponds to the increase in density and, consequently, the compacted structure with low porosity [46].

The air permeance value was 1.29 s/100 cm$^3$ for the KP sample, but after the deposition of the nanocellulose coatings, the maximum flow time of the air column, 1800 s, was extrapolated for all samples. This air permeability measurement was also extrapolated for

the F1 and F2 films, showing that the resistances to air passage of the films and papers with coatings were higher than the range measured by this method, indicating their low permeability to air [24]. Heat-based sterilization treatments, such as pasteurization, require packaging made from airtight materials to maintain their microbiocidal effect [47], as airtight packaging protects against external contamination and cross-contamination between foods stored in the same compartment, in addition to preserving flavor and food texture [3].

When assessing the water permeability of paper, coating papers, and films through their absorption kinetics, the contact angle decreased over time. In the F1 and F2 films, there was a decrease in the contact angle of water in contact with an oxygen atmosphere; however, after 15 s of contact with the surface, it was not possible to measure the angle formed, and droplet spreading occurred rather than water absorption. For the control, the same behavior was observed, except that the droplets were quickly absorbed by the paper. These results are confirmed by the water absorption analysis shown in Table 2, where the water absorption of the F1 and F2 films was very low compared to that of KP.

**Table 2.** Water absorption kinetics through contact angle analysis.

| Sample | 0 s | 5 s'' | 15 s | 30 s |
|---|---|---|---|---|
| KP | 65.36 d (11.79) | 18.45 a (30.28) | 0 | 0 |
| K1 cs | 53.37 cd (32.75) | 45.03 bc (35.42) | 43.47 a (46.08) | 44.28 a (49.34) |
| K1 ps | 64.38 d (16.19) | 0 | 0 | 0 |
| K2 cs | 45.46 bc (17.09) | 41.18 bc (23.82) | 42.32 a (27.59) | 41.95 a (28.32) |
| K2 ps | 64.42 d (12.32) | 0 | 0 | 0 |
| F1 | 36.14 ab (14.06) | 25.43 ab (20.86) | 0 | 0 |
| F2 | 26.58 a (32.82) | 20.75 a (16.70) | 0 | 0 |

cs: coating side. ps: paper side. Means followed by the same letters indicate no statistical differences by the Tukey test at 95% probability. Values in parentheses refer to the coefficient of variation in percent.

For the K1 and K2 samples, when analyzed from the side where the NFC coating was deposited, the contact angle remained practically constant throughout the adopted kinetics. For the same samples analyzed on the side opposite the NFC coating, the absorption was practically instantaneous. This result reaffirms the result found for the water absorption, where the absorption on this surface was higher than that on KP and three times higher than the absorption on the surface of the same sample when analyzed on the NFC coating side.

Because of its higher viscosity, it was more difficult for glycerol to penetrate the pores, and a decrease in the contact angle was observed with increasing contact time for all samples (Table 3). For samples F1, F2, K1 cs, and K2 cs, surface spreading was observed, whereas for samples KP, K1 ps, and K2 ps, droplet absorption occurred, causing the paper to swell.

**Table 3.** Glycerol absorption kinetics through contact angle analysis.

| Sample | 0 s | 5 s'' | 15 s | 30 s |
|---|---|---|---|---|
| KP | 86.56 cd (9.55) | 49.01 b (13.47) | 35.88 b (18.70) | 27.92 cd (19.27) |
| K1 cs | 71.24 a (12.60) | 42.79 ab (34.89) | 29.80 ab (31.90) | 22.90 bc (21.64) |
| K1 ps | 92.20 d (5.98) | 51.70 b (15.40) | 36.57 b (18.51) | 30.56 cd (22.40) |
| K2 cs | 80.33 abcd (13.98) | 46.51 ab (25.99) | 35.57 b (36.99) | 27.58 cd (20.99) |
| K2 ps | 82.03 bcd (9.18) | 45.54 ab (19.09) | 34.53 b (26.83) | 33.21 d (20.67) |
| F1 | 77.71 abc (11.98) | 34.48 a (25.07) | 20.41 a (29.86) | 11.64 a (33.46) |
| F2 | 74.06 ab (8.22) | 35.30 a (14.82) | 23.47 a (15.33) | 15.60 ab (24.84) |

cs: coating side. ps: paper side. Means followed by the same letters indicate no statistical differences by the Tukey test at 95% probability. Values in parentheses refer to the coefficient of variation in percent.

Generally, smoother surfaces have lower contact angles than rougher surfaces [48]. The structures of the NFC films and coatings are significantly more compact; therefore, less water and glycerol can penetrate them compared to pure kraft paper samples [49].

### 3.3. Mechanical Tests

The mechanical properties were significantly influenced by the addition of nanocellulose coatings (Table 4).

**Table 4.** Mechanical properties.

| Sample | BI (kPa·m$^2$/g) | RI (mN·m$^2$/g) | TI (N·m/g) |
|--------|------------------|-----------------|-------------|
| KP | 4.20 a [(5.76)] | 18.00 a [(3.80)] | 57.80 a [(10.27)] |
| K1 | 5.56 b [(20.99)] | 19.57 b [(2.62)] | 67.24 ab [(7.64)] |
| K2 | 7.42 c [(10.53)] | 21.40 c [(2.60)] | 75.42 bc [(8.58)] |
| F1 | 2.16 d [(14.84)] | - | 83.07 c [(8.27)] |
| F2 | 3.18 e [(28.93)] | - | 76.05 bc [(18.06)] |

BI: burst index. RI: tear index. TI: tensile index. Means followed by the same letters indicate no statistical differences by the Tukey test at 95% probability. Values in parentheses refer to the coefficient of variation in percent.

KP showed a lower burst index (BI) than the samples with K1 and K2 surface depositions, with gains of 31.38% and 76.67%, respectively. However, the films had a lower burst index than the KP, K1, and K2 samples.

For the tear index (RI), the increase in relation to the CS after the depositions was lower than that for the burst index, being 8.72% and 18.9% for K1 and K2, respectively.

The tensile index (TI) was the highest for the films, and the increases in resistance in relation to KP were 43.7% and 31.6% for F1 and F2, respectively. After the depositions, the percentage increases were 16.33% and 30.5% for K1 and K2, respectively.

When using nanocellulose, the expected increase in tensile and bursting strength is related to the degree of fiber interlacing and fiber length [24], whereas the tearing effect of fiber interlacing is less pronounced because it is a property that strongly depends on the thickness of the cell wall [24,50].

For the films (F1 and F2), the TI remained high even without paper as a base, showing the effect of fibrillation on its increase [24]. However, there was a drop in BI, indicating that it also depends on the characteristics of the fiber (cell wall thickness and fiber length) that make it up and not solely on the degree of fibrillation [24,51]. The RI is zero for the films because the cell walls of the fibers that make them up are very thin owing to the fibrillation process by which they were produced [24].

### 3.4. Thermal Stability

The thermal degradation of the samples began after 200 °C (Table 5). There is greater stability for the KP samples in relation to those with a deposition up to 360 °C (Figure 4), where the percentage loss after this temperature is lower for the coating papers and for the film.

**Table 5.** Mass loss in thermal ranges.

| Sample | Mass Loss (%) | | | | |
|--------|---------------|---|---|---|---|
| | 100–200 °C | 200–300 °C | 300–400 °C | 400–500 °C | Residual Mass (%) |
| KP | 0.38 | 4.15 | 65.11 | 3.26 | 27.10 |
| K1 | 0.75 | 4.16 | 63.13 | 3.95 | 28.02 |
| K2 | 0.56 | 4.18 | 65.99 | 4.20 | 25.06 |
| F1 and F2 | 0.02 | 6.47 | 56.98 | 3.85 | 32.69 |

The NFC exhibited the highest mass loss up to 300 °C, but in the 300–400 °C range, the cellulose pulp showed the lowest thermal stability. The nanocellulose films exhibited the lowest thermal degradation after 500 °C, maintaining a residual content of over 30%. The greater bonding between the fibrils increased the thermal stability of the films [24].

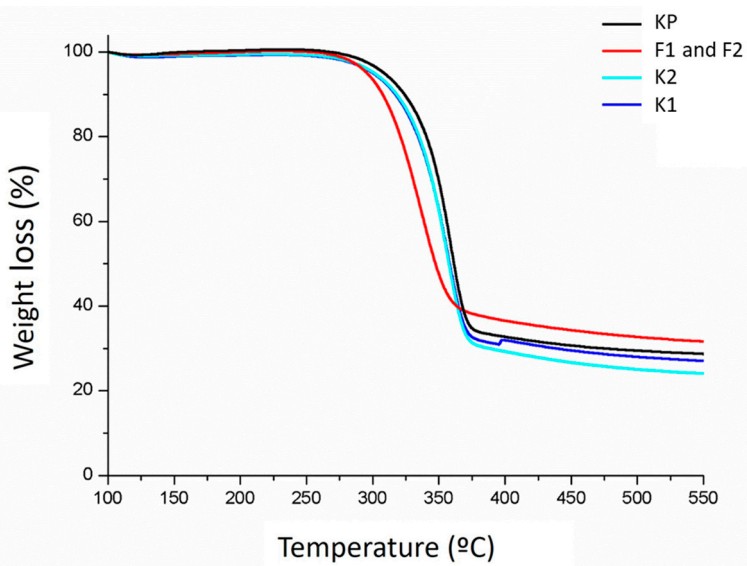

**Figure 4.** Mass loss with increasing temperature.

The thermal stability was not influenced by the thickness of the film used for the surface coating. All samples behaved similarly to KP because of their similar chemical compositions. The lower thermal stability of the nanocellulose film was due to the bleaching of the sample, which allowed the cellulose to break down more quickly [52] (Table 6). The maximum and final degradation temperatures of the nanocellulose film were lower than those of the other samples, owing to the absence of lignin in the film [52]. However, all samples behaved similarly in terms of the start, maximum, and end temperatures of degradation.

**Table 6.** Temperatures of the thermal degradation intervals of the samples.

| Sample | Thermal Degradation Intervals (°C) | | |
|---|---|---|---|
| | $T_{onset}$ | $T_{max}$ | $T_{endset}$ |
| KP | 226 | 359 | 390 |
| K1 | 221 | 357 | 390 |
| K2 | 237 | 357 | 394 |
| F1 and F2 | 232 | 341 | 283 |

$T_{onset}$: temperature at the onset of thermal decomposition. $T_{max}$: temperature where the maximum mass variation occurs due to degradation. $T_{endset}$: temperature at the end of thermal decomposition.

To be used in food packaging, materials must withstand temperatures ranging from −40 °C to 220 °C [53]. Therefore, all the treatments explored fulfilled this premise. Comparatively, the $T_{onset}$ results are similar to those of PP (215–242 °C) and lower than those of PS (260–280 °C) and PET (330–390 °C) [54]. $T_{max}$ and $T_{endset}$ are also similar to those of PP ($T_{max}$: 293–353 °C and $T_{endset}$: 346–381 °C) and lower than those of PS ($T_{max}$: 392–412 °C and $T_{endset}$: 424–433 °C) and PET ($T_{max}$: 431 °C and $T_{endset}$: 484 °C) [54].

## 4. Conclusions

As shown above, based on the analysis results of the morphological, physical, mechanical, and thermal properties, the use of nanofibrillated cellulose (NFC) as a coating can improve the performance of kraft paper for food packaging. With the application of the coatings, the porosity of the kraft papers decreased and the density increased. The water absorption on the coating side decreased, whereas that on the paper side increased.

The bulk density, water absorption and final dry thickness were not affected by the deposition thickness of the NFC suspension.

The NFC films absorbed less water, and the different thicknesses did not influence the apparent density or water absorption. Uncoated kraft paper was the only air-permeable material.

The NFC coatings improved the mechanical properties of the paper, with a coating thickness of 2 mm increasing the bursting resistance and tear resistance and a coating thickness of 1 mm improving the tensile strength.

All the evaluated materials demonstrated the minimum thermal stability required in the production of food packaging, with the onset of thermal degradation occurring at temperatures greater than 220 °C.

**Author Contributions:** Conceptualization, E.C.L., L.C.S. and G.I.B.d.M.; methodology, E.C.L.; formal analysis, A.S.d.A., A.N.L. and E.L.S.d.M.L.; investigation, E.C.L. and E.A.B.J.; resources, E.C.L. and E.A.B.J.; data curation, E.C.L. and E.A.B.J.; writing—original draft preparation, E.C.L. and E.A.B.J.; writing—review and editing, E.C.L. and E.A.B.J.; visualization, A.S.d.A.; supervision, E.C.L., L.C.S. and G.I.B.d.M.; project administration, E.C.L.; funding acquisition, E.C.L. and E.A.B.J. All authors have read and agreed to the published version of the manuscript.

**Funding:** This study was financed by Research Support Foundation of the State of Mato Grosso (FAPEMAT) (FAPEMAT.0000077/2022: Notice No. 010/2021 Research with a High Level of Technological Maturity PANMT).

**Institutional Review Board Statement:** Not applicable.

**Informed Consent Statement:** Not applicable.

**Data Availability Statement:** Data are contained within the article.

**Acknowledgments:** The authors are grateful to the Pulp and Paper Laboratory team from Federal University of Paraná for all the support in this paper and to the Electron Microscopy Center (CME-UFPR).

**Conflicts of Interest:** The authors declare no conflict of interest.

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
