# Peer review of "Nanocellulose Coating on Kraft Paper"

_coatings, doi:10.3390/coatings13101705_

Round 1

Reviewer 1 Report

In my opinion, the manuscript could have the potential to fit within the scope of the journal only after a major revision. As a matter of fact, it seems, in my opinion, that there are many points in the manuscript to clarify/improve: clarify the novelty, improve the literature, add the amount of NFC/Kraft paper used and improve the description of the thermal properties.

Line 14: “Paper…environment.” In my opinion, the sentence is not clear. I advise writing, “Paper is a biodegradable material, but, in food packaging, its hygroscopicity and porosity can cause food contamination due to the exchange of gasses and liquids with the environment.”

Line 17: change “is” with “are”. I advise adding “less hygroscopic and porous.”

Line 48: There are biodegradable plastics. I suggest writing something like “The most used plastics are not biodegradable…”. At the same time, using biodegradable plastic is not necessarily the solution to pollution. I suggest this review article: Current Research in Green and Sustainable Chemistry 5 (2022) 100273, https://doi.org/10.1016/j.crgsc.2022.100273

Line 60: I advise adding literature for the different pulping processes. A suggestion: Tofani, G., Cornet, I. & Tavernier, S. Estimation of hydrogen peroxide effectivity during bleaching using the Kappa number. Chem. Pap. 75, 5749–5758 (2021). https://doi.org/10.1007/s11696-021-01756-y

Line 95: Can the authors explain what is the novelty of their work? The use of Nanocellulose as a coating agent is already known.

Line 106: How much NFC was used to make the films (g, mg…)? What was the mass of kraft paper used in the preparation of the samples?

Line 251: The addition of some photos of the contact angle can boost the quality of the paper.

Line 312:  In my opinion, Table 5 is not clear. Please clarify. For example, F1 and F2 show 32.69 % mass loss, higher than other samples, but It is not coherent with Figure 5 and the text, line 319, “lowest thermal degradation after 500°”)

Line 314: What are CS and Filme? Please use the acronyms used until now (KP, F1 and F2).

From my side, I have found only one main problem in the first sentence of the abstract.

Author Response

Many of the revisions were accepted, however some suggestions were subjective, and we justified the reason for not accepting them.

Reviewer 2 Report

The presented manuscript examines issues related to the production of new packaging materials that can be used primarily for packaging products. The main goal is to replace packaging made from synthetic polymers, which are often not recyclable. To impart functional properties to paper packaging, it is necessary to create an appropriate layer on the surface of the paper. Such layers should be made of biodegradable materials. In this case, I recommend that authors add examples that are already known and presented in the literature.
For example, in the work of Khomutinnikov, N.V. et al, https://doi.org/10.3390/polym15030692 it is proposed to form such layers by treating with a direct solvent - N-Methylmorpholine-N-oxide cellulose, which is non-toxic and is regenerated a large number of times.
In their literature review, the authors note the high demand for this type of packaging materials. The volume of products in the future may reach millions of tons.
In their work, the authors show that applying an NFC layer to paper can reduce moisture sorption by 55-58%.
Lines 92, 93. It’s not entirely clear whether the thickness of the paper or the surface layer is indicated?
Line 104. Perhaps not microfibrils, but nanofibrils?!
2.2. Coating of papers and films production.
I'm confused by the thickness of the layer, perhaps the authors mean microns rather than mm?!
Lines 223-229. But it’s completely clear, first the authors talk about a decrease in sorption, and then about an increase for samples with and without coating?!
It is necessary to add a scale bar to Figures 1, 3; without it it is difficult to estimate the size of the fibrils.
How do the authors explain the insignificant values of mass loss associated with the loss of sorbed water?
Why does thermal degradation start earlier for the Filme sample? If the reason for this is the characteristics of the objects used, then this must be shown!
How did the authors control the thickness of the created layer? Can the authors provide microphotographs of the transverse cleavage?

The quality of the figures needs to be improved. The manuscript contains typos and inaccuracies, for example, in lines 157, Figure 5, etc.

The manuscript contains typos and inaccuracies, for example, in lines 157, Figure 5, etc.

Author Response

We accepted almost all suggestions.

Reviewer 3 Report

The manuscript concerns the use of nanocellulose coatings on kraft paper.

the manuscript raises a very interesting topic with great application potential

The planned research and their discussion are appropriate.

The manuscript can be published in its present form

Only in the case of thermal stability tests would I consider additional TGA in air.

Some minor comments are listed below:

-          Line 66; Na2S, no subscript.

-          lines 154 and 158; units should be corrected, dot symbol

-          line 175; space before the comma, please remove

-          Tables 1, 2, 3, 4 and 5; a comma instead of a period for values in the tables.

-          Figure 5; markings inconsistent with the previous ones, (CS, Filme), please correct.

Author Response

It was a very valuable review and pointed out several small errors that went unnoticed by us. We were unable to make just the recommendation about the TGA.

Reviewer 4 Report

The study is devoted to optimizing the properties of paper packaging for food storage, which is undoubtedly of interest.

The results of the work concluded that untreated kraft paper was the only breathable material, and post-treatment reduces breathability but improves mechanical properties (increases tear resistance and increases tear strength).

The article does not discuss in any way whether the fact that the treated paper becomes airtight will greatly affect the storage conditions of the products. In my opinion, this can significantly affect the storage of products. I recommend adding a discussion on this part, maybe humidity measurements during storage.

It would also be appropriate to add to the discussion why cellulose was considered as a coating for paper, what materials could be used in addition to it.

Author Response

This reviewer asked for two points in the discussion, one of which we accepted, the other was already covered in the introduction.

Round 2

Reviewer 1 Report

In my opinion, the article does not provide sufficient novelty for publication. The use of nanocellulose as a coating agent is already known, and, in my opinion, the authors did not provide adequate clarification for the novelty of the manuscript either in the paper or in the reply.

Also, the experimental part is still not adequately described. It is not a problem to not report the weight of nanocellulose and Kraft paper and use the consistency of nanocellulose and the grammage. However, I did not find the volumes of paper and nanocellulose used in the "Materials and Methods". In my opinion, it is not possible to repeat the experiment.

Moreover, the authors reported in the reply to the reviewer and the first version of the manuscript the use of 1 % nanocellulose suspension. However, in the new version of the manuscript, the nanocellulose suspension become 1.3 % without being marked in yellow as modification.

Also, the authors accepted the use of " The most used plastics are not biodegradable…”. However, the sentence is still the same in the manuscript (line 48). The same in the abstract, the modifications were accepted, but the manuscript was not modified.

Author Response

The answers are in the attached document.

Reviewer 2 Report

The authors responded to previous comments and questions, and additional ones are provided below:
Fig.1 "apers" typo
Figure 2. I recommend correcting the figure caption to avoid repetitions.
Lines 227-235. Perhaps the process of water removal occurs unevenly because nanocellulose has a more perfect structure. As a result, not only a change in density should occur, but also in the morphology of the paper surface towards a greater number of defects and inhomogeneities, including deviations in thickness. Can the authors comment on these assumptions?
Line 241. "absorption on the surface" I may enter "adsorption on the surface" here.
Figure 4. The authors need to pay attention to the curve for sample “K1”; why does the mass increase at 400°C?

Fig.1 "apers" typo

Author Response

(The authors gave the same response as above.)

Reviewer 4 Report

The authors answered my questions sufficiently, overall I am satisfied

Author Response

We are grateful for reviews.

Round 3

Reviewer 1 Report

The authors clarified all the comments. In my opinion, the manuscript can be accepted.